# Mechanisms of A-Type Lamin Targeting to Nuclear Ruptures Are Disrupted in *LMNA-* and *BANF1-*Associated Progerias

**DOI:** 10.3390/cells11050865

**Published:** 2022-03-02

**Authors:** Rhiannon M. Sears, Kyle J. Roux

**Affiliations:** 1Enabling Technologies Group, Sanford Research, Sioux Falls, SD 57104, USA; rhiannon.sears@sanfordhealth.org; 2Basic Biomedical Sciences, Sanford School of Medicine, University of South Dakota, Vermillion, SD 57069, USA; 3Department of Pediatrics, Sanford School of Medicine, University of South Dakota, Sioux Falls, SD 57104, USA

**Keywords:** nuclear envelope, nuclear rupture, progeria, *LMNA*, lamin, HGPS, BAF, *BANF1*, NGPS

## Abstract

Mutations in the genes *LMNA* and *BANF1* can lead to accelerated aging syndromes called progeria. The protein products of these genes, A-type lamins and BAF, respectively, are nuclear envelope (NE) proteins that interact and participate in various cellular processes, including nuclear envelope rupture and repair. BAF localizes to sites of nuclear rupture and recruits NE-repair machinery, including the LEM-domain proteins, ESCRT-III complex, A-type lamins, and membranes. Here, we show that it is a mobile, nucleoplasmic population of A-type lamins that is rapidly recruited to ruptures in a BAF-dependent manner via BAF’s association with the Ig-like β fold domain of A-type lamins. These initially mobile lamins become progressively stabilized at the site of rupture. Farnesylated prelamin A and lamin B1 fail to localize to nuclear ruptures, unless that farnesylation is inhibited. Progeria-associated *LMNA* mutations inhibit the recruitment affected A-type lamin to nuclear ruptures, due to either permanent farnesylation or inhibition of BAF binding. A progeria-associated BAF mutant targets to nuclear ruptures but is unable to recruit A-type lamins. Together, these data reveal the mechanisms that determine how lamins respond to nuclear ruptures and how progeric mutations of *LMNA* and *BANF1* impair recruitment of A-type lamins to nuclear ruptures.

## 1. Introduction

The nuclear envelope (NE) surrounds the nucleus during interphase to functionally compartmentalize the cell, enable various signaling and regulatory processes, and protect the genome. A specialized extension of the endoplasmic reticulum (ER), the NE has two connected phospholipid bilayers that form the outer nuclear membrane (ONM) and inner nuclear membrane (INM) [1]. In Metazoa, a meshwork of type-V intermediate filaments known as the nuclear lamina is connected to the INM. Composed of individual lamins each in their individual fiber meshwork [2], the nuclear lamina provides structural support to the nucleus, interacts with and retains numerous resident INM transmembrane (TM) proteins and helps anchor the peripheral heterochromatin [3,4]. In mammals, A-type lamins, lamin A (LaA) and lamin C (LaC), are encoded from the *LMNA* gene via alternative splicing; whereas the B-type lamins, lamin B1 (LaB1) and lamin B2 (LaB2) are encoded on separate genes, *LMNB1* and *LMNB2,* respectively. All lamins share three distinct domains, a rod domain composed of multiple coiled-coil domains, a nuclear localization sequence (NLS), and an immunoglobulin-like β fold (hereafter denoted Ig-like fold) domain [3,5]. Additionally, the lamina interacts with the protein complex that spans the NE to connect the nucleoskeleton to the cytoskeleton (LINC complex) [6]. Nuclear pore complexes (NPCs) are large proteinaceous channels that sit within annular holes within the NE at sites where the ONM and INM are connected. The NPCs regulate transport between the nucleoplasm and cytoplasm [7] and are mechanically integrated within the nuclear lamina [8,9,10,11,12].

The structure and function of the NE exists only during interphase and undergoes regulated disassembly early in mitosis and reassembly during mitotic exit [13]. However, during interphase the NE can become compromised and rupture, exposing the nuclear compartment to cytosolic constituents. Instances of nuclear rupture have been observed in both natural and pathological circumstances, including during cellular migration or compression [14,15,16,17,18,19,20]. Nuclear rupture has also been implicated in several diseases, such as cancer [14,21,22,23], autoimmunity [24,25,26], and in diseases of the lamina, called laminopathies [27,28,29,30,31,32]. Though most nuclear ruptures are thought to undergo quick and efficient repair [14,15,33,34,35], evidence suggests that transient nuclear ruptures can lead to DNA damage [23,36,37,38,39,40], senescence [25,26], and altered transcription [29,40].

There are over 600 mutations in the *LMNA* gene that are associated with as many as 15 different laminopathies (reviewed in [41,42]). One of these diseases is Hutchinson-Gilford progeria syndrome (HGPS), with symptoms partially resembling accelerated aging. The vast majority of HGPS is caused by a de novo autosomal dominant point mutation in *LMNA* that results in a splicing defect of pre-lamin A (preLaA) and the accumulation of a preLaA protein lacking 50 amino acids called progerin, or LaA Δ50 [43,44]. Under normal conditions, preLaA, LaB1, and LaB2 undergo the posttranslational addition of a farnesyl group via farnesyltransferase to the Cys residue of a C-terminal CaaX motif [45]. Subsequently, the zinc metalloprotease ZmpSte24 or RCE1 cleaves the C-terminal -aaX tripeptide [46,47,48] and isoprenylcysteine carboxymethyl transferase (ICMT) carboxymethylates the newly generated C-terminus [49]. B-type lamins remain permanently farnesylated/carboxymethylated [5], but preLaA undergoes one final ZmpSte24 cleavage 14 amino acids upstream of the C-terminus to generate a mature LaA that lacks the C-terminal farnesylation [50,51,52]. The internally truncated progerin protein is missing the final ZmpSte24 cleavage site and remains permanently farnesylated [53]. Less frequently, various missense mutations in the *LNMA* gene can also cause HGPS or HGPS-like diseases without impacting the maturation process of LaA [53,54]. 

Other progeroid syndromes that arise from mutations in NE-associated proteins include Nestor-Guillermo progeria syndrome (NGPS), caused by a recessive single missense mutation in the small, soluble DNA binding protein barrier-to-autointegration factor (BAF; *BANF1*) [55,56,57], and LEMD2-associated progeria syndrome (LPS), resulting from an autosomal dominant missense mutation in a TM INM protein called LEM (LAP2, Emerin, MAN1) domain-containing protein 2 (LEMD2; *LEMD2*) [58]. Although NGPS and LPS are both less severe forms of progeria than HGPS, these diseases share some phenotypic consequences, including at the cellular level misshapen nuclei and mislocalization of nuclear proteins [53,56,57,58,59]. These three progeria-associated NE proteins, LaA, BAF, and LEMD2, are all involved in nuclear structure and organization [60,61,62,63] and have been implicated in both post-mitotic NE reformation [62,64,65,66] and in repair of nuclear ruptures [15,17,67,68]. Evidence suggests that these three proteins may coexist in a molecular complex where BAF dimers bind to the LEM domain of various LEM-domain proteins, including LEMD2, and to the Ig-like fold of A-type lamins, potentially acting as a bridge between LEMD2 and LaA to function synergistically to accomplish one or more cellular functions [69]. Progeria syndromes caused by mutations in these proteins may result from disrupted interactions of these proteins within the complex [63,69] or with other cellular constituents that interact with these proteins to accomplish a shared function [57,66,67,70,71].

Recent evidence suggests nuclear rupture may be an underlying mechanism of the vascular pathology seen in a HGPS mouse model, in which ruptures were identified in aortic smooth muscle cells preceding the loss of these cells in the HGPS mice [72]. Furthermore, perturbation of the LINC complex in this HGPS mouse model alleviated the loss of the aortic smooth muscle cells by disrupting the cytoskeletal force transmission to the nucleus [73]. Although *LMNA* mutations have been associated with an increased predisposition to nuclear rupture [28,29,30,31,32] and lamin A has been shown to accumulate at nuclear ruptures [14,15,68], a thorough study of lamin behavior at nuclear ruptures remains unreported. 

Here, we revealed how lamins behave in response to nuclear rupture and the mechanisms underlying that behavior. We further determined how progeric mutations in *LMNA* and *BANF1* disrupt targeting of A-type lamins to nuclear ruptures, potentially contributing to the underlying mechanism(s) of disease. 

## 2. Materials and Methods

### 2.1. Cell Culture

HEK293T Phoenix (National Gene Vector Biorepository, Indianapolis, IN, USA) and NIH3T3 (ATCC; CRL-1658) cell lines were cultured in DMEM with 4.5 g/L glucose, L-glutamine, and sodium pyruvate (Corning, Glendale, AZ, USA) and supplemented with 10% (*v*/*v*) fetal bovine serum (FBS; Hyclone, Logan, UT, USA). BJ-5ta cell lines were cultured in DMEM supplemented with a 4:1 ratio of Medium 199 (Sigma-Aldrich, St. Louis, MO, USA) and 0.01 mg/mL hygromycin B (Invitrogen, Carlsbad, CA, USA) and 10% (*v*/*v*) FBS. MCF10A (ATCC; CRL-10317) cell lines were cultured in Mammary Epithelium Basal Medium (Lonza, Walkerville, MD, USA) supplemented with MEGM SingleQuots (Lonza) with the following modification: gentamicin sulfate-amphotericin was omitted from media, and cholera toxin (Sigma-Aldrich) was added at a final concentration of 100 ng/mL. All cells were grown at 37 °C with 5% CO_2_ in a humidified incubator. 

### 2.2. Plasmids

All plasmids used were generated using the In-Fusion cloning system (Takara, San Jose, CA, USA) with primers containing 15 bp 5′ flanking regions complementary to the free ends of the cloning vector. All plasmids contain human cDNA. Specific primer sequences are provided in Appendix A. Enhanced GFP was used in all GFP-tagged constructs. GFP-LaA WT, L647R, L530P, K542N, and Δ50 were PCR amplified from human templates using forward (fwd) primer 1593 and reverse (rev) primer 1651 and inserted into mycBioID2 pBabe puro (Addgene #80901) cut EcoRI to PmeI. GFP-LaC was amplified from a human template via fwd primer 1593 and rev primer 2036 and inserted into mycBioID2 pBabe puro cut EcoRI to PmeI. GFP-LaB1 was amplified from a human template using fwd primer 1593 and rev primer 2035 and inserted into mycBioID2 pBabe puro cut EcoRI to PmeI. Fwd primer 1593 and rev primer 2105 amplified the GFP-tagged LaA rod (aa 1–435) and was inserted into mycBioID2 pBabe puro cut EcoRI to PmeI. To create the GFP-LaA tail, the LaA tail (aa 391–664) was amplified via fwd primer 2104 and rev primer 1651 and inserted into GFP-LaA WT pBabe puro cut XhoI-Pme1. The Δ50 and K542N tails were made the same way. Amplified LaA via fwd primer 1873 and rev primer 1874 and amplified LaA tail via fwd primer 2181 and rev primer 1874 were inserted into mCherry-NLS pBabe neo [67] cut BspE1 to Sal1. Enzymatically inactive human cGAS^E225A/D227A^-mCherry (denoted cGAS-mCherry throughout manuscript; a generous gift from Jan Lammerding) was amplified using fwd primer 1452 and rev primer 1453 and inserted into a pBabe neo vetor cut EcoRI to SalI. The siRNA-resistant BAF WT was synthesized at Gene Universal (Newark, DE, USA) using codon optimization and inserted into pBabe puro cut BamHI to SalI. To make the siRNA-resistant A12T the WT construct was amplified with fwd primer 1927 containing the point mutation and rev primer 1928 and inserted into pBabe puro cut BamH1 to SalI. GFP-siRNA-resistant BAF was made by amplifying BAF WT or A12 T with fwd primer 2004 and reverse primer 2005 and inserted into our previously described GFP-BAF pBabe puro [67] cut XhoI to SalI. To make the bicistronic vector siRNA-resistant BAF WT or A12T-IRES-GFP-NLSx3 pBabe puro construct, fwd primer 1924 and rev primer 1925 amplified either BAF WT or BAF A12T. The internal ribosome entry site (IRES) was amplified with fwd primer 1998 and reverse primer 1903 from pMSCV-IRES-mCherry-FP (Addgene #52114). GFP-NLSx3 was amplified from our previously described GFP-NLSx3 pBabe puro vector [67] using fwd primer 1833 and rev primer 1834. The three PCR amplifications were then annealed, recombined, and inserted into pBabe puro cut SnaBI to SalI.

### 2.3. Stable Cell Line Generation

All fluorescent fusion proteins were stably expressed in cell lines generated by retroviral transduction. HEK293 Phoenix cells were transfected with the pBabe plasmid DNA encoding for the protein of interest using Lipofectamine 3000 (Thermo Fisher Scientific Waltham, MA, USA) per manufacturer’s recommendation. Transfected Phoenix cells were incubated at 37 °C overnight, then replenished with fresh culture medium and transferred to 32 °C for 24 h. The culture media were collected and filtered through individual 0.45-μm filters. Filtered medium was added to the corresponding target cells with polybrene (2.5 μg/mL; Santa Cruz Biotechnology Dallas, TX, USA), and target cells were incubated at 37 °C for 48 h. Target cells were then incubated in fresh culture media containing either puromycin (0.5 μg/mL; Thermo Fisher Scientific) for 2–3 d for pBabe puro plasmid or G418 sulfate (30 μg/mL; Santa Cruz Biotechnology) for 5–7 d for pBabe neo plasmids for selection of viral integration. Expression of fusion proteins in cell lines was further verified using immunofluorescence and immunoblot analysis. 

### 2.4. siRNA Transfection

All siRNA transfections were performed using RNAiMAX (Thermo Fisher Scientific) according to the manufacturer’s recommendations. Cells were seeded at 70% confluency into 6-well tissue-culture plates containing 2 mL DMEM and allowed to adhere to the bottom of the plate during an overnight incubation at 37 °C. Cells were then treated with siRNA oligos for 96–120 h. ON-TARGETplus SMARTpool siRNAs (Dharmacon, Lafayette, CO, USA) against mouse LaA/C (NM_001002011.3), human BAF (NM_003860.3), human LEMD2 (NM_181336.4), human Emerin (NM_000117.2), human Ankle2 (NM_015114.3) were used for gene knockdowns, including a nontargeting control (Appendix A). For knockdowns of mouse siLaA/C in NIH3T3 cells, the siRNA oligos were transfected twice during the first 72 h of the 120-h incubations. Double transfections of *Silencer* select validated siRNA (Ambion, Austin, TX, USA) against human ZmpSte24 (NM_005857.3) was used for ZmpSte24 depletion for 144 h (Appendix A). For live-cell experiments, 48 h before imaging cells were split onto 35-mm glass-plated FluoroDishes (World Precision Instruments, Sarasota, FL, USA) along with a parallel 6-well plate for collecting cell lysates for immunoblot verification of siRNA knockdown efficiency via immunofluorescence and/or immunoblot. 

### 2.5. Immunofluorescence 

Cells grown on 1.5 mm glass coverslips were fixed with 3% (wt/vol) paraformaldehyde/phosphate-buffered saline (PFA/PBS) for 10 min and permeabilized by 0.4% (wt/vol) Triton X-100/PBS for 15 min before endogenous proteins were identified through indirect fluorescence. Cells were labeled for 1 h in primary antibodies: goat anti-preLaA (1:50; sc-6214, Santa Cruz Biotechnology) mouse anti-LaA/C (1:100; 1E4; provided by Dr. Frank McKeon), rabbit anti-LaA/C pS22 (1:100; 2026S; Cell Signaling Technology, Danvers, MA, USA), rabbit anti-BAF (1:100; ab129184; Abcam, Cambridge, UK), and mouse anti-BAF (1:100; H00008815-M07; Abnova, Taipei, Taiwan). Primary antibodies were detected using Alexa Flour 568-conjugated donkey anti-goat (1:1000; A11057; Thermo Fisher Scientific), Alexa Flour 568-conjugated goat anti-rabbit (1:1000; A11011; Thermo Fisher Scientific), Alexa Flour 568-conjugated goat anti-mouse (1:1000; A11004; Thermo Fisher Scientific), Alexa Flour 647-conjugated goat anti-rabbit (1:1000; A21244; Thermo Fisher Scientific), Alexa Flour 647-conjugated goat anti-mouse (1:1000; A21235; Thermo Fisher Scientific) and Hoescht dye 33,342 to detect DNA. Coverslips were mounted using 10% (wt/vol) Mowiol 4–88 (Polysciences, Inc., Warrington, PA, USA). Epifluorescent images were captured using a Nikon Eclipse NiE (40×/0.75 Plan Fluor Nikon objective; 20×/0.75 Plan Apo Nikon objective) microscope at room temperature with a charge-coupled device camera (CoolSnap HQ; Photometrics, Tucson, AZ, USA) linked to a workstation running NIS-Elements software (Nikon, Melville, NY, USA). All images were processed in Adobe Photoshop CC 2017 (Adobe, San Jose, CA, USA) for cropping and brightness/contrast adjustment when applicable.

### 2.6. Immunoblotting

Total cell lysates were used to analyze protein expression. 1.2 × 10^6^ cells were lysed using SDS-PAGE sample buffer, boiled for 5 min, and sonicated to shear DNA. Proteins were separated on 4–20% gradient (Mini-PROTEAN TGX; Bio-Rad, Hercules, CA, USA) and transferred to nitrocellulose membrane (Bio-Rad). Membranes were blocked with 10% (vol/vol) adult bovine serum and 0.2% Triton X-100 in PBS for 30 min, and then incubated with appropriate primary antibodies: rabbit anti-LaA/C (1:1000; 2032S; Cell Signaling Technology), rabbit anti-GFP (1:1000; ab290; Abcam), rabbit anti–BAF (1:1000; ab129184; Abcam), rabbit anti–LEMD2 (1:1000; HPA017340; Atlas Antibodies, Bromma, Sweden), rabbit anti-Emerin (1:1000; 2659S; Cell Signaling Technology), and rabbit anti-Ankle2 (1:1000; ab225905; Cell Signaling Technology). Mouse monoclonal anti-tubulin (1:5000; sc-32293; Santa Cruz Biotechnology) was used as a loading control. The primary antibodies were detected using horseradish peroxidase (HRP)–conjugated anti-rabbit (1:5000; G21234; Thermo Fisher Scientific) or anti-mouse (1:5000; F21453; Thermo Fisher Scientific) antibodies. The signals from antibodies were detected using enhanced chemiluminescence via a LI-COR ODYSSEY Fc Imaging system (LI-COR Biosciences, Lincoln, NE, USA).

### 2.7. Laser-Induced Nuclear Rupture and Live Cell Imaging

Live cells expressing fluorescently-tagged proteins of interest were seeded onto 35-mm glass-bottom FluoroDishes in DMEM 48 h before imaging. The day of imaging, the media was removed and replaced with pre-warmed phenol red–free DMEM with Hepes and FBS (Gibco, Grand Island, NY, USA) for imaging. Cells were imaged on an Olympus FV1000 confocal microscope and FV10-ASW v4.1 software (Olympus Corporation, Tokyo, Japan), with a temperature-controlled chamber set at 37 °C and 60×/NA 1.42 Plan Apo N oil immersion objective. GFP and mCherry imaging was completed via the 488-nm and 543-nm scanning lasers, respectively. Laser-induced nuclear rupturing was performed by focusing the 405-nm excitation laser at 100% power (tornado scan mode) in a small ROI on the nuclear envelope for 6–8 s. Utilizing the SIM-scan feature allowed for simultaneous imaging and laser-induced rupturing. The rupture reporter cGAS-mCherry was used as a secondary rupture reporter in all cells unless otherwise indicated. GFP-LaA, GFP-LaC, and GFP-BAF nuclear photobleaching was performed using the main scanner and a 488-nm laser at 100% power for 10 s or until no noticeable signal was observed in the nucleoplasmic compartment. All images were processed in Photoshop CC 2017 (Adobe) for cropping and brightness/contrast adjustments when applicable. Videos were made by exporting AVIs from the FV10-ASW v4.1 software and importing them into Windows Movie Maker v8.0.7.5 (Microsoft, Redmond, WA, USA).

### 2.8. FTI Treatments

Farnesyl transferase inhibitor-277 (FTI-277) (Sigma-Aldrich) was prepared as a stock at 10 mM. Cells were incubated in media containing 10 µM FTI-277 for 48 h before experimentation. For cells being treated with both FTI-277 and siRNA transfections, cells underwent siRNA transfection 48 h before they started treatment with FTI-277 for 48 h before live cell imaging. 

### 2.9. Quantification of Proteins during Nuclear Rupture

To measure GFP-tagged lamin intensity at the nuclear rupture site, we measured the intensity of GFP-lamins at the rupture site 5 min post rupture. All values were normalized to the fluorescence of a non-ruptured region on the lamina opposite to the nuclear rupture at 5 min post rupture. GFP-BAF WT or A12T average intensity was measured at 3 min post rupture at the rupture site, at a distal location from the rupture site in the nucleoplasm, and on the nuclear envelope adjacent to the rupture. Each measurement was normalized to the average GFP-BAF intensity of the total cell at 3 min post rupture. All intensity measurements were mean intensity measurements and performed in ImageJ v1.52i (National Institutes of Health, Bethesda, MD, USA).

### 2.10. GFP-LaA Stabilization at Rupture

To measure lamin A intensity at the nuclear rupture site overtime, we measured the intensity of GFP-LaA at the rupture site before rupture, during the 405-nm laser tornado scan, and every minute after the tornado scan for 15 min. All values were normalized to the fluorescence of a region on the lamina adjacent to the nuclear rupture at the corresponding time points. For assessment of lamin A stabilization into the lamina at the rupture site, a 488-nm laser at 100% power was used to bleach one half of the GFP-LaA accumulated at the rupture site for 5 s at either 2, 5, 10, or 15 min post rupture. GFP-LaA changes at the rupture site were recorded for 2 min after the half-rupture bleach. Quantification of GFP-LaA intensity at rupture sites was performed 2 min after half-rupture bleaches at 2, 5, 10, and 15 min by measuring the intensity at each time point before the 5 s half-rupture bleach and again 2 min later in either the bleached half or unbleached half of the rupture. The ratio of the 2 min post half-rupture bleach to before the half-rupture bleach was calculated for each the bleached half and unbleached half of the rupture. The adjacent lamina intensity was calculated at 15 min by calculating the ratio before the half-rupture bleach to 2 min post half-rupture bleach. All intensity measurements were mean intensity measurements and performed in ImageJ v1.52i (National Institutes of Health) by measuring the mean intensity of interest of a region of interest at designated time points. All values were normalized to the fluorescence of a region on the lamina on the opposite side of the nucleus from the nuclear rupture at the corresponding time points.

### 2.11. Population Cell Compressions

MCF10A cells were grown on 1.5 mm glass coverslips in a 24 well plate with 0.5 mL media 24 h before cell compression to induce mechanical nuclear ruptures. To induce mechanical stress, the coverslips were placed cells face-up into the bottom 35-mm glass-bottom FluoroDish set in a 1-well static confiner cell compression device (4Dcell, Montreuil, France) and directly compressed using a Polydimethylsiloxane (PDMS) confinement piston for 5 s. The compression device allowed for an equally distributed force to be generated on the cells through the PDMS confinement piston. Cells were placed back into the 24 well plate with warm media and incubated at 37 °C and 5% CO_2_ for the indicated time periods before fixing and labeling for immunofluorescence.

### 2.12. Statistical Analysis

A one-way ANOVA followed by a Tukey’s post hoc multiple comparison test was used for analysis on GFP-tagged lamin intensity at the nuclear rupture site and cGAS-mCherry diameter at 5 min post rupture. To analyze the GFP-LaA stabilization at the rupture site over time, a two-way ANOVA followed by a Tukey’s post hoc multiple comparison test was used to evaluate the relationship of GFP-LaA intensity within the unbleached-half and bleached-half of the LaA accumulation at rupture at each time point. Unpaired student’s *t* test were performed both to compare nuclear preLaA relative fluorescence intensity between siControl and siZmpSte24 treatments and to compare GFP-BAF WT vs. A12T dynamics 3 min post rupture. Significance was determined if *p* < 0.05. All graphs represent mean values ± SEM (error bars) unless otherwise denoted. Statistical analysis and graph generation were performed in GraphPad Prism v.7.02 (GraphPad Software Inc., San Diego, CA, USA). 

## 3. Results

### 3.1. Comparison of Lamin Behaviors during Nuclear Ruptures 

To observe lamin behavior in real-time during the process of nuclear rupture and repair, GFP-tagged human LaA, LaC, or LaB1 (Figure 1A) were stably expressed in BJ-5ta human fibroblasts along with the established nuclear rupture marker cGAS-mCherry, a predominantly cytoplasmic protein that binds to newly exposed genomic DNA at the rupture site [14,15,67]. Upon laser-induced nuclear rupture [67] of these cells at a single, defined site on the nuclear envelope (Figure 1B), we observed rapid recruitment and enrichment of GFP-LaA and GFP-LaC but not GFP-LaB1 at the rupture sites that persisted for >10 min (Figure 1B, Appendix A), despite the nuclear ruptures being similar in size (Appendix A). To explore the dynamic accumulation and potential stabilization of LaA at nuclear ruptures, we compared the intensity of GFP-LaA at the rupture site over time compared to an adjacent non-ruptured region of the NE. The accumulation of GFP-LaA occurs quickly and reaches maximum intensity at ~10 min following rupture at almost 2-fold excess of that normally localized at the envelope. After 10 min, the levels of GFP-LaA at the rupture site began to decrease suggestive that this accumulation gradually returns to normal levels (Figure 1C). For LaC we observed an even more profound accumulation at the nuclear ruptures (Appendix A).

Since we observed persistent (>10 min) bleached regions on the NE on either sides of the GFP-LaA accumulation, outside of the initial cGAS-mCherry-defined nuclear rupture boundaries (Appendix A), we hypothesized that it was likely the more mobile, nucleoplasmic population of LaA [74,75,76] that is being recruited to the rupture sites, and the adjacent photobleached lamina retains its normal relative immobility [74,77]. To demonstrate if it was indeed the mobile population of A-type lamins that localizes to the ruptures, the nucleoplasmic pool of GFP-tagged A-type lamins was photobleached prior to rupture. This resulted in a loss of GFP-LaA accumulation at the rupture and greatly diminished the amount of the more mobile [75,78] GFP-LaC (Figure 1D). To visualize endogenous mobile A-type lamin behavior during a nuclear rupture, we mechanically-induced nuclear rupture on MCF10A cells via a cell compression chamber (Appendix A) and stained with 1E4, a mouse monoclonal antibody that preferentially recognizes nucleoplasmic A-type lamins [79]. We also utilized an antibody that detects A-type lamins phosphorylated at serine-22 (LaA/C pS22), normally generated during early mitotic dissolution of the lamina, but also found in the nucleoplasm during interphase [78]. We found the mobile, nucleoplasmic pool of endogenous A-type lamins accumulate at rupture sites (Appendix A), and A-type lamins targeting to the rupture site are phosphorylated at Ser-22 (Appendix A). 

To determine if and when newly recruited LaA becomes stabilized in the lamina at the rupture site, at 2, 5, 10, or 15 min post-rupture we photobleached the recruited LaA within one half of the rupture site and visualized the changes in GFP-LaA intensity at the bleached site for 2 min (Figure 1E). As expected from our prior studies (Figure 1C), when photobleached at the 2, 5, and 10 min time points we observed a gradually decreasing accumulation of GFP-LaA at the bleached portion of the rupture site that cannot be explained by a concomitant loss from the unbleached portion of the rupture site within 2 min after the half-rupture bleach (Figure 1F). By 2 min after the 15 min post-rupture bleach, we can observe an inability to substantially recruit new GFP-LaA to the bleached portion of the rupture (Figure 1F, *, *p* = 0.0298) suggesting a stabilization of the GFP-LaA within the lamina. A region of the NE adjacent to the rupture was also measured to assess the overall photobleaching during the experiment (Figure 1F, red bar). Together, these results support that mobile, nucleoplasmic A-type lamins are recruited to nuclear ruptures in excess of levels normally found at the nuclear envelope and a subpopulation of these lamins are gradually stabilized within the nuclear lamina. 

### 3.2. Farnesylated Lamins Do Not Accumulate at Ruptures

We hypothesized that LaB1 was not recruited to nuclear ruptures (Figure 1B) either due to farnesylation that may inhibit nucleoplasmic localization, and thus mobility, or due to an inability to interact with retention mechanisms at the site of rupture. We tested the first hypothesis indirectly by investigating if permanent farnesylation of LaA would inhibit its localization to ruptures. To induce preLaA accumulation in BJ-5ta fibroblasts coexpressing GFP-LaA and cGAS-mCherry, we used siRNA to knockdown ZmpSte24, the final enzyme required to cleave the farnesyl group from preLaA during LaA post-translational maturation [51,52]. Verification of ZmpSte24 depletion was performed via immunofluorescence detection of preLaA (Appendix A). Depletion of ZmpSte24 significantly increased the nuclear preLaA relative intensity when compared to control (Appendix A) and resulted in loss of GFP-LaA accumulation at rupture sites (Figure 2A, Appendix A). To verify this observation, we expressed a GFP-tagged LaA L647R, an atypical progeria mutant [80] that inhibits the final ZmpSte24 cleavage of the farnesylated tail of preLaA [81]. The permanently farnesylated LaA L647R did not accumulate at rupture sites (Figure 2B and Appendix A). Furthermore, we treated the GFP-LaA L647R with 10 μM farnesyl transferase inhibitor (FTI)-277 for 48 h to prevent the initial addition of the farnesyl group to the prelamin’s CaaX domain during the post translational modification process of lamin A [82]. After treatment with FTI-277 for 48 h, GFP-LaA L647R accumulated at ruptures (Figure 2C and Appendix A) similarly to wildtype LaA (Appendix A). We also observed GFP-LaB1 recruitment to ruptures after FTI-277 treatment (Figure 2D and Appendix A). By 5 min post rupture, levels of FTI-treated GFP-LaB1 accumulation at the rupture site were similar to that of the non-ruptured lamina (Appendix A). This data further supports that the farnesylation of lamins inhibit their recruitment to nuclear ruptures, and not that LaB1 inherently lacks an ability to interact with mechanisms of lamin recruitment at rupture sites.

### 3.3. A-Type Lamin Targeting to Nuclear Ruptures

To determine which domain(s) within LaA is required for recruitment to rupture sites, we stably expressed human GFP-tagged LaA or domain-deletions in mouse NIH3T3 fibroblast cells and depleted the endogenous mouse A-type lamins via siRNA (Appendix A). Full-length (FL) GFP-LaA displayed normal accumulation at the rupture sites upon depletion of endogenous mouse LaA/C (Figure 3A and Appendix A). Expression of the LaA rod domain (aa 1–435) revealed a similar localization to the FL LaA at the nuclear envelope prior to rupture, but the LaA rod failed to accumulate at nuclear ruptures (Figure 3B and Appendix A). The LaA tail (aa 391–646) was exclusively nucleoplasmic prior to ruptures and localized at rupture sites (Figure 3C and Appendix A). The LaA tail contains the Ig-like fold; therefore, we hypothesized that this structure may be responsible for targeting LaA to rupture sites. To test this, in human BJ-5ta fibroblasts we expressed a GFP-tagged LaA L530P mutant predicted to destabilize the protein folding of the Ig-like fold [83] but that otherwise localizes normally in the nucleus [77]. The GFP-LaA L530P failed to accumulate at rupture sites (Figure 3D), suggesting that the Ig-like fold is critical for the targeting of LaA to ruptures. Previously, we have shown that BAF is required to recruit LEM-domain proteins, rupture repair proteins, and membranes to the site of nuclear ruptures [67]. BAF has been shown to be necessary for the recruitment of LaA to sites of rupture [68], and BAF is known to interact with the Ig-like fold of LaA [69]. To demonstrate if BAF is responsible for recruiting not only LaA, but also LaC, the LaA tail, and non-farnesylated LaB1 to nuclear rupture sites, we depleted BAF prior to laser-induced rupture (Figure 3E). Indeed, all observed lamin recruitment to nuclear ruptures, including the Ig-like fold of LaA and nonfarnesylated LaB1 is BAF-dependent (Figure 3E and Appendix A). To verify that loss of LaA recruitment was due to BAF depletion and not lack of nuclear membrane recruitment to the rupture site, we depleted three transmembrane LEM-domain proteins LEMD2, Ankle2, and emerin that were previously shown to cause a similar defect in the repair of nuclear ruptures when compared with BAF depletion [67]. GFP-LaA accumulated to rupture sites normally when these three LEM-domain proteins were depleted (Appendix A). Confirmation of endogenous protein depletion was performed via immunoblot (Appendix A).

### 3.4. Progeric Lamin Behaviors during Nuclear Ruptures

Dominant mutations in *LMNA* that lead to permanent farnesylation of LaA occur in the vast majority of HGPS patients [53]. Activation of a cryptic splice site leads to the generation of a LaA isoform that lacks 50 amino acids (LaA Δ50) within the C-terminus which removes the final ZmpSte24 cleavage site but retains the CaaX motif needed for farnesylation [53]. Less common mutations in the Ig-fold of A-type lamins, including the recessive K542N mutation on the surface of the Ig-fold [54], can also lead to HGPS phenotypes [84,85,86,87,88]. We sought to assess how these progeric mutations impact the behavior of LaA in response to nuclear rupture. Stable expression of GFP-tagged LaA WT, LaA Δ50, or LaA K542N, in BJ-5ta fibroblasts revealed that neither progeric variant accumulates properly at rupture sites (Figure 4A and Appendix A). We then treated these cells with 10 μM FTI-277 for 48 h before rupture and observed that GFP-LaA Δ50 accumulated at rupture sites (Figure 4B and Appendix A), indicating permanent farnesylation was the likely cause of preventing this accumulation. FTI treatment did not result in K542N recruitment to nuclear ruptures (Figure 4B). Furthermore, expression of LaA Δ50 did not inhibit wildtype LaA accumulation at ruptures (Figure 4C). We subsequently assessed the behavior of the progeric LaA tails during nuclear rupture. In NIH3T3 cells expressing the GFP-LaA Δ50 tail and depleted of the endogenous LaA/C, we observed a partial envelope association as well as accumulation and enrichment of the protein at the rupture sites that persisted for at least 10 min (Figure 4D and Appendix A). After a 48 h FTI-277 treatment in cells expressing the LaA Δ50 tail, we observed a shift to a higher molecular mass of the GFP-LaA Δ50 tail via immunoblot (Figure 4E), indicating the LaA Δ50 tail is farnesylated and explaining the partial envelope association. The LaA K542N tail, however, did not target to the rupture site in cells depleted of their endogenous A-type lamins (Figure 4D and Appendix A). Collectively, this data suggests although both progeric mutations prevent protein accumulation at nuclear rupture sites, the mechanisms underlying the loss of recruitment are different. The ability of the more mobile LaA Δ50 tail but not the less mobile full-length LaA Δ50 to localize to nuclear ruptures suggests that it is not the inability of the LaA tail to interact with BAF when farnesylated (Figure 4E) that prevents targeting of the full-length LaA Δ50 but the lack of a mobile population of a permanently farnesylated LaA [70,71,89]. LaA K542N, however, lacks a strong Ig-like fold association with BAF [69], and is thus unable to localize to nuclear ruptures even when artificially mobilized by removal of the stabilizing coiled-coil domain.

### 3.5. Lamin Recruitment to Ruptures in the Presence of Progeric BAF 

Since BAF recruits lamins to the nuclear rupture, likely due to direct interaction at least in the case of A-type lamins, and progeric mutations in *LMNA* prevent targeting to ruptures either by loss of mobility or an inability to interact with BAF, we sought to ascertain if the NGPS mutation of BAF also prevents recruitment of A-type lamins to nuclear ruptures. Due to BAF being essential for proper mitotic exit [62], we developed a siRNA-resistant version of GFP-BAF WT and the NGPS recessive mutant GFP-BAF A12T, allowing for selective depletion of endogenous BAF via siRNA (Appendix A). At steady-state, GFP-BAF WT stably expressed in BJ-5ta fibroblasts exhibits a cytoplasmic, nuclear, and very strong NE population (Figure 5A); whereas GFP-BAF A12T displays a diminished nuclear envelope population and greater cytoplasmic and nucleoplasmic populations (Figure 5B), likely due to a loss of interaction with the lamina [69]. To visualize BAF’s recruitment to the rupture and subsequent influx into the nucleoplasm, we bleached the nuclear compartment of the cells immediately before laser-induced nuclear rupture. Cytosolic BAF WT robustly aggregates at the site of the nuclear rupture, before slowly dissociating from the DNA at the rupture site and onto the NE and into the nucleoplasm (Figure 5A and Appendix A; [67]). Cytosolic BAF A12T also aggregates at the rupture site, but then quickly diffuses into the nucleoplasm without strongly localizing at the NE (Figure 5B and Appendix A). By 3 min post nuclear rupture, less BAF A12T had accumulated at the rupture site than WT (Figure 5C,D; **, *p* < 0.0001) and more BAF A12T diffused into the nucleoplasm opposite the rupture site (Figure 5C,E; **, *p* < 0.0001). BAF A12T also demonstrated a significant reduction in NE association at an adjacent, non-ruptured location near the rupture site when compared to BAF WT 3 min post nuclear rupture (Figure 5C,F; *, *p* < 0.05). 

Since we observed the diminished NE population of BAF A12T and since the mutation is located at the BAF-lamin Ig-like fold binding interface [69], we sought if BAF A12T was capable of recruiting LaA to nuclear rupture sites. We stably expressed an untagged version of our siRNA-resistant BAF WT or A12T in a bicistronic vector that co-expresses a GFP-NLS nuclear rupture marker via an internal ribosome entry site (IRES). We then stably coexpressed either a mCherry-LaA or mCherry-LaA tail in these cells. Following endogenous BAF depletion (Appendix A) and laser-induced nuclear rupture, BAF WT but not BAF A12T was able to recruit both LaA and the LaA tail (Figure 5G,H and Appendix A), indicating any recruitment we saw in the siControl cells expressing BAF A12T was due to the interaction of LaA with endogenous BAF. Collectively, these studies support that BAF recruits A-type lamins to nuclear ruptures via an interaction with the lamin Ig-like fold and that the recessive NGPS mutation of *BANF1* reduces the nuclear envelope association of BAF both before and during nuclear rupture and disrupts A-type lamin recruitment to nuclear ruptures. 

## 4. Discussion

These studies have revealed the mechanisms that mediate lamin behavior during interphase rupture of the NE and how progeria-associated mutations in *LMNA* and *BANF1* inhibit that process. A preexisting population of mobile, and at least partially Ser-22 phosphorylated, nucleoplasmic A-type lamins rapidly accumulates at rupture sites due to an interaction with BAF and begin to appreciably stabilize at the rupture site after ~10–15 min, suggestive of a functional repair of the nuclear lamina, likely involving dephosphorylation by phosphatases to stabilize filament formation [89]. This repair of the lamina is a process that would be expected to occur to prevent subsequent ruptures due to a weakened site in the structural scaffold. Farnesylation inhibits lamin recruitment to rupture sites, thus only mature-LaA and LaC substantially participate in this process. However, both preLaA, LaA Δ50, and LaB1 can accumulate at ruptures if farnesylation is inhibited, indicating that this hydrophobic motif is somehow preventing these lamins from reaching the rupture and binding to BAF. Further clarity for this inhibitory mechanism of farnesylation comes from the successful recruitment to nuclear ruptures of the permanently farnesylated LaA Δ50 tail. This C-terminal portion of LaA lacks the stabilizing coiled-coil rod domain and remains substantially mobile in the nucleoplasm, indicating that the farnesylation itself is not inhibitory to the interaction unless in combination with the stabilizing coiled-coil domain needed to retain the lamins in the lamina. Collectively, these results would suggest that cells expressing A-type lamins have a mechanism to repair the nuclear lamina relatively rapidly at sites of nuclear ruptures to inhibit subsequent ruptures due to a focal weakening or deficiency of the structural lamina. In cells that do not express A-type lamins, such a mechanism to repair the lamina would presumably be lacking as the farnesylated B-type lamins would be immobile. Indeed, levels of A-type lamins increase during development, especially in cells exposed to mechanical forces that may induce nuclear ruptures [90], and these lamins not only act to resist forces and prevent rupture [91] but could participate in the rapid repair of the lamina following rupture. 

Previously, it has been shown that cytosolic BAF binds to the genomic DNA exposed following a nuclear rupture and is responsible for recruitment of transmembrane LEM-domain proteins and their associated nuclear membranes to functionally repair the NE [67,68]. Lamin A recruitment to ruptures was also found to be BAF-dependent [68]. Here, we demonstrated BAF is capable of recruiting A-type lamins to ruptures via the Ig-like fold. Furthermore, if artificially nonfarnesylated, LaB1 can also be recruited to nuclear ruptures in a BAF dependent manner. There is no evidence for a nonfarnesylated population of LaB1, so this is likely irrelevant for naturally occurring nuclear ruptures, but it does suggest that BAF can also bind to B-type lamins, directly or indirectly. The direct interaction of the Ig-like fold of A-type lamins with BAF is inhibited by the K542N substitution in the Ig-like fold [69] and results in failure to be recruited to nuclear ruptures. Similarly, the A12T mutation is predicted to impact the BAF-LaA/C binding interface, has a decreased association on the NE prior to and during rupture, and is unable to recruit LaA to nuclear ruptures. Together, these results reinforce the model that BAF directly binds to the Ig-like fold of LaA and perturbation of that interaction inhibits recruitment of A-type lamins to nuclear ruptures.

It is tempting to suggest that since progeric LaA, either LaA Δ50, K542N or L647R and progeric BAF A12T all share a defect in lamin recruitment to nuclear ruptures that this may be an underlying mechanism of disease. This concept could be further supported by the recent studies suggesting that nuclear ruptures do pathologically occur in a mouse model of progeria [72]. However, the situation is clearly more nuanced and complicated. The predominant *LMNA* mutation that causes HGPS is an autosomal dominant activation of a cryptic splice site that affects considerably less than 100% of the transcripts from the mutated allele [92,93] and the mutation only impacts splicing of LaA and not LaC. Thus, the level of progeric LaA Δ50 is relatively low compared to the total level of A-type lamins, being substantially less than 50% in most cells. We also demonstrated here that LaA 50 expression does not inhibit LaA WT accumulation at the rupture sites. The similarly farnesylated autosomal dominant L647R mutation, while developed as a tool to study lamin-A processing by ZmpSte24 [50], leads to a progeria with generally less severe symptoms than observed in conventional HGPS patients despite being fully penetrant for the one impacted allele [80]. On the other hand, the autosomal recessive K542N mutation affects both alleles and all isoforms with 100% of the A-type lamins being impacted and unable to be recruited to sites of nuclear rupture. However, the disease severity is not appreciably worse, albeit with few patients having been reported and from a single family [54]. It is perhaps noteworthy that cells from these patients are reported to have frequent blebs deficient in A- and B-type lamins but with accumulations of LEM domain proteins indicative of potential nuclear ruptures [94]. It is also noteworthy that the autosomal dominant *LMNA* EDMD mutation L530P similarly impacts both *LMNA* protein isoforms and fails to localize to the nuclear ruptures despite being more mobile than WT [77], but in contrast with K542N does not lead to progeria. It can be assumed that the complete loss of the Ig-like fold structure and all of its associated interactions, including the LINC complex proteins Sun1/2 [95] and emerin [96], caused by the L530P mutation would constitute a loss of function, albeit only impacting one allele, whereas the K542N mutation may only impact interaction with BAF or at least a subset of interactions of the Ig-like fold. It is also possible that the mechanisms that underlie the progeria-associated deficiency in recruitment of A-type lamins to nuclear ruptures is indicative of defects in the BAF-dependent process of nuclear envelope reformation following mitosis which could underlie disease phenotypes. 

What remains unclear from the studies presented here are the consequences of recruiting A-type lamins to nuclear ruptures or, conversely, failing to do so. It is known that loss of A-type lamins can lead to more severe leakage from the nucleus during induced ruptures [67]. However, this is as much likely due to intrinsic changes in the mechanics of the lamina as in the recruitment of lamins to the rupture. We speculate that this recruitment helps in the repair of the nuclear lamina and thus promotes integrity of the nuclear envelope in cells with a prior intrinsic or extrinsic force that led to the initial rupture and may thus be susceptible to subsequent ruptures perhaps even forming a reinforced ‘scar’ at the rupture site. However, the stabilization of these proteins at the rupture site takes considerably longer than is required for the functional repair of these ruptures [67] suggesting that it may not be involved in the functional repair itself. Studies to clarify this will be challenging since inhibiting lamin localization to the nuclear rupture also disturbs other interphase functions of the lamins or the BAF that recruits them. 

## Figures and Tables

**Figure 1 cells-11-00865-f001:**
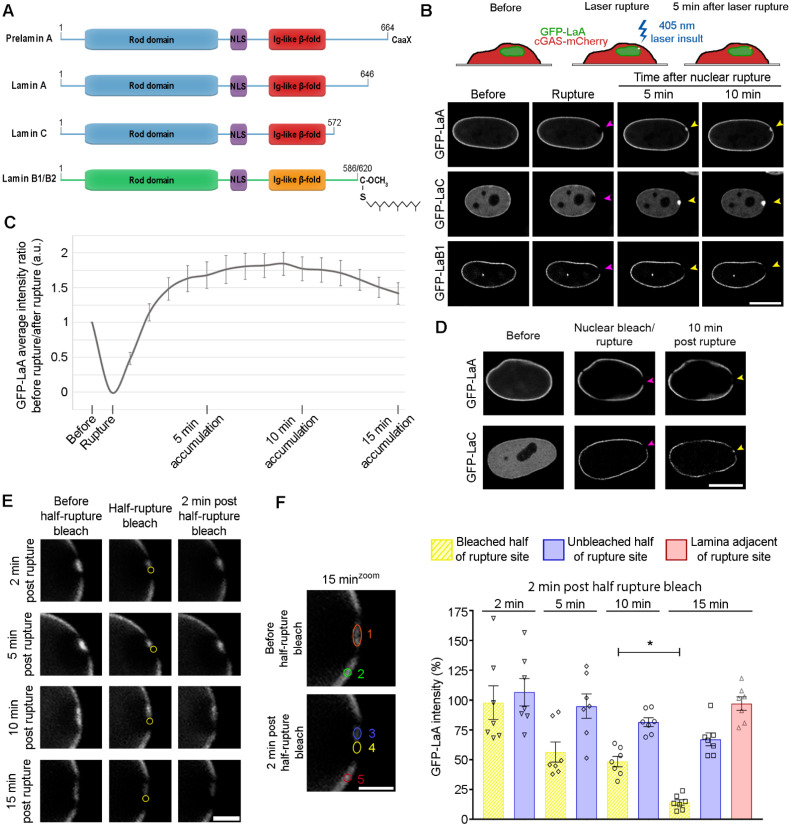
Mobile A-type lamins accumulate at nuclear ruptures and gradually become immobile. (**A**) Linear models of lamins that highlight important features of the proteins, including the coiled-coil rod domain, the nuclear localization sequence (NLS), and the Ig-like β-fold. B-type lamins and preLaA are farnesylated, whereas mature LaA and LaC are not. (**B**) A 405 laser is used to induce nuclear rupture at a precise location on the nuclear envelope. BJ-5ta cells stably expressing GFP-tagged LaA, LaC, or LaB1 after a laser-induced nuclear rupture (purple arrowheads) were imaged for 10 min to monitor for protein accumulation at the rupture sites (yellow arrowheads). Scale bar, 10 μm (**C**) Graphical representation of the ratio of GFP-LaA intensity prior to and following the laser-induced nuclear rupture in stably expressing BJ-5ta cells. All values are normalized to an adjacent site on the nuclear envelope to account for photobleaching during image capture. The graph represents mean values ± SEM (*n* = 8 cells). (**D**) Representative images of BJ-5ta cells expressing either GFP-LaA or GFP-LaC had their nucleoplasmic GFP signal photobleached, then underwent laser-induced nuclear rupture (purple arrowheads) and were imaged for 10 min to assess protein accumulation (yellow arrowheads). Scale bar, 10 μm. (**E**) GFP-LaA mobility at ruptures sites was determined at time points 2, 5, 10, and 15 min post laser-induced rupture by bleaching half of the rupture for 5 s (yellow circle), then waiting 2 min before measuring changes in GFP-LaA intensity within the two rupture halves. Scale bar, 2 μm. (**F**) Quantification of GFP-LaA intensity at rupture sites were performed 2 min after half-rupture bleaches at 2, 5, 10 or 15 min. The average intensity of LaA was measured at each time point before a 5 s half-rupture bleach (orange circle 1) and again 2 min later in either the bleached half (yellow circle 4) or unbleached half (blue circle 3) of the rupture. The ratio of the 2 min post half-rupture bleach to before the half-rupture bleach was calculated and shown as a percentage for each the bleached half and unbleached half of the rupture. The adjacent lamina intensity at 15 min was determined by calculating the ratio before the half-rupture bleach (green circle 2) to 2 min post half-rupture bleach (red circle 5). All values were normalized to the fluorescence of a region on the lamina on the opposite side of the nucleus from the nuclear rupture at the corresponding time points. The graph represents mean values ± SEM and includes individual values (*n* = 7 cells for each time point; *, *p* = 0.0298 by a two-way ANOVA with Tukey’s post hoc multiple comparison test).

**Figure 2 cells-11-00865-f002:**
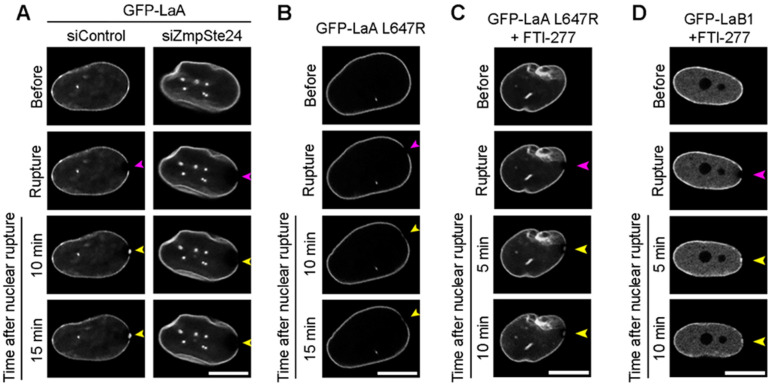
Farnesylated lamins do not accumulate at nuclear ruptures but inhibiting farnesylation promotes rupture accumulation. Representative images of laser-induced nuclear rupture events in BJ-5ta cells stably expressing (**A**) GFP-LaA that underwent either control or ZmpSte24 siRNA knockdown, (**B**) GFP-LaA L647R, (**C**) FTI-treated GFP-LaA L647R, or (**D**) FTI-treated GFP-LaB1. All cells were imaged for at least 10 min after laser-induced nuclear rupture (purple arrowheads) to monitor for protein accumulation (yellow arrowheads). Scale bars, 10 μm.

**Figure 3 cells-11-00865-f003:**
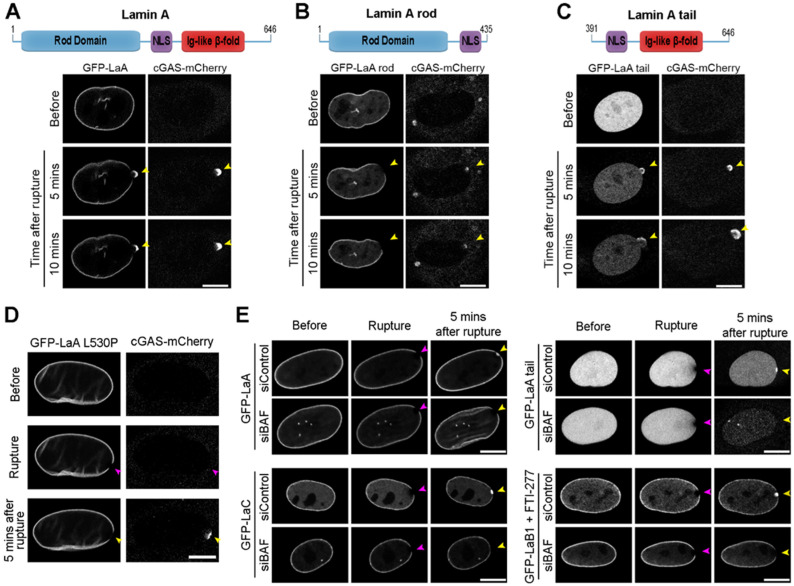
Lamin recruitment to nuclear ruptures is dependent on the lamin Ig-fold domain and BAF. NIH3T3 mouse fibroblast cells stably co-expressing cGAS-mCherry and GFP-tagged human (**A**) full-length LaA, (**B**) LaA rod (aa 1–435), or (**C**) LaA tail (aa 391–646) were depleted of endogenous A-type lamins by siRNA prior to laser-induced nuclear rupture. GFP-fusion protein accumulation at the rupture site was monitored for 10 min. Purple arrowheads indicate initial site of laser application. Yellow arrowheads indicate location of nuclear rupture. (**D**) BJ-5ta cells stably overexpressing GFP-LaA L530P and cGAS-mCherry were subjected to laser-induced nuclear rupture. (**E**) Representative images of BJ-5ta cells stably expressing GFP-LaA, GFP-LaC, GFP-LaA tail, or FTI-277-treated GFP-LaB1 that underwent laser-induced nuclear rupture following siRNA control or BAF knockdown. Scale bars, 10 μm.

**Figure 4 cells-11-00865-f004:**
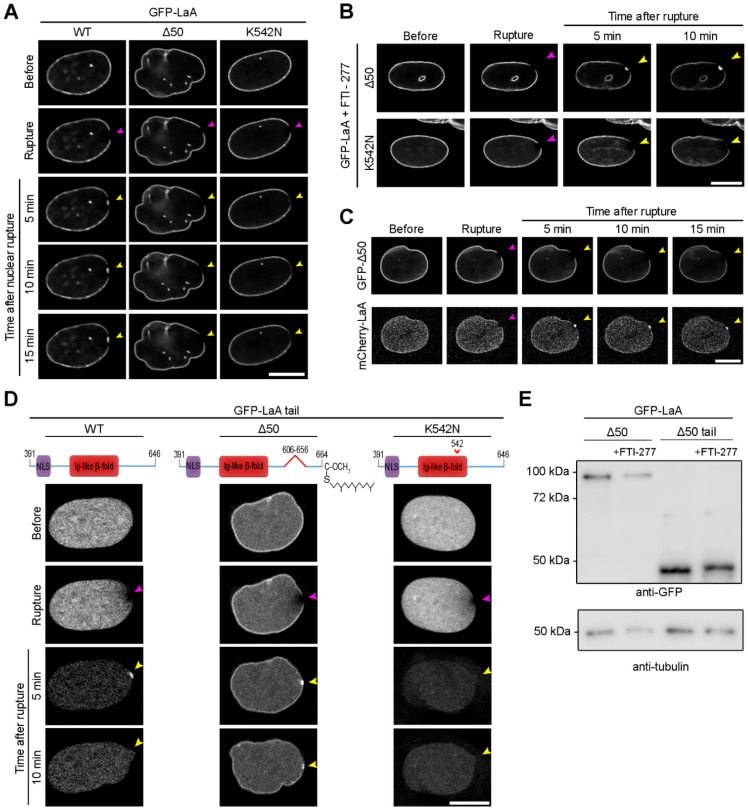
Progeric lamin A Δ50 and K542N fail to localize to nuclear ruptures by different mechanisms. (**A**) Representative images of BJ-5ta cells stably expressing either GFP-LaA WT, Δ50, or K542N prior to and following laser-induced nuclear rupture. Rupture sites were monitored for lamin accumulation for 15 min. Purple arrowheads indicate site of laser-induced rupture. Yellow arrowheads indicate location of expected protein accumulation. (**B**) BJ-5ta cells expressing either GFP-LaA Δ50 or GFP-LaA K542N were treated with FTI-277 before rupture via laser ablation and monitored for lamin accumulation at rupture sites for 10 min. (**C**) BJ-5ta cells stably co-expressing GFP-LaA Δ50 and mCherry-LaA WT were monitored 15 min after laser-induced nuclear rupture for lamin accumulation at the rupture site. (**D**) NIH3T3 cells stably expressing human GFP-LaA WT tail, Δ50 tail, or K542N tail were depleted of the endogenous LaA/C via siRNA before laser induced rupture. Lamin accumulation at the rupture site was monitored over 10 min after rupture. (**E**) Immunoblot analysis of cells expressing the permanently farnesylated GFP-LaA Δ50 and GFP-LaA Δ50 tail with or without FTI-277 treatment. The increased molecular weight in the FTI-treated samples indicates inhibition of farnesylation and subsequent CaaX processing. Anti-tubulin was used as a protein loading control. Scale bars, 10 μm.

**Figure 5 cells-11-00865-f005:**
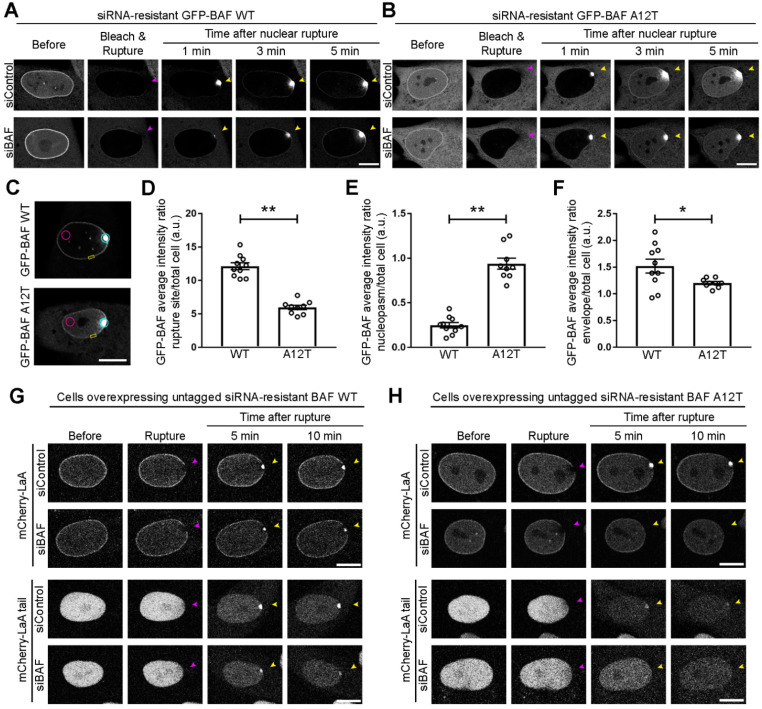
Progeric BAF A12T exhibits reduced NE-association prior to and during rupture and is unable to recruit lamin A to nuclear ruptures. BJ-5ta cells expressing a GFP-tagged siRNA-resistant human BAF (**A**) WT or (**B**) A12T underwent endogenous BAF depletion via siRNA transfection prior to nuclear compartment photobleaching and laser-induced nuclear rupture. Migration of GFP-BAF into the nucleus was monitored for 5 min. Purple arrowheads indicate site of laser-induced rupture. Yellow arrowheads indicate location of expected protein accumulation. (**C**) Representative location of regions of interest (ROI) for quantification at 3 min post rupture of the average GFP-BAF intensity at the rupture site (blue ROI), nucleoplasm distal to rupture site (pink ROI), and nuclear envelope (NE) (yellow ROI). The average GFP-BAF intensity ratio of the (**D**) rupture site, (**E**) the nucleoplasm distal from the rupture site, or (**F**) a site on the nuclear envelope adjacent to the rupture was compared to average total cell intensity of GFP-BAF at 3 min post rupture. The graph represents mean values ± SEM and includes individual values (*n* = 10 cells for WT and 9 cells for A12T, ** *p* < 0.0001 and * *p* < 0.05 by an unpaired student’s *t* test). BJ-5ta cells stably co-expressing expressing GFP-NLS and untagged siRNA-resistant BAF (**G**) WT or (**H**) A12T via an internal ribosomal entry site (IRES) were co-expressed with either mCherry-LaA or mCherry-LaA tail and depleted of endogenous BAF via siRNA transfection prior to laser-induced nuclear rupture. Lamin accumulation at the rupture site was monitored over 10 min following rupture. Scale bars, 10 μm.

## Data Availability

There were no data deposited in publicly-available data repositories for this project.

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
