# Peer review of "Mechanisms of A-Type Lamin Targeting to Nuclear Ruptures Are Disrupted in LMNA- and BANF1-Associated Progerias"

_cells, 2022, doi:10.3390/cells11050865_

Round 1

Reviewer 1 Report

The authors have submitted a well written manuscript describing the recruitment of nucleoplasmic lamins A and C to sites of nuclear envelope rupture and the role of lamin farnesylation and interaction with BAF in the process. The experiments are carefully designed and nicely support the authors well balanced conclusions, and the images and videos are stunning and of high quality. Thus, the results will be of interest to a broad community. Nonetheless, the authors should address a number of items to further strengthen the manuscript and its findings.

  • The main concern is that many of the findings, with the exception of data presented in Figure 1D and F, are rather qualitative. Although the images and image sequences are convincing, it is difficult to assess how strong and robust the differences are. The authors should provide information on the number of cells analyzed. In addition, the authors should include quantification of key findings in their other figures (Fig. 2-4), for example, measurements of the fluorescent intensity of the GFP-tagged proteins at the site of nuclear envelope rupture, normalized to the intensity at other nuclear envelope sites, along with appropriate statistical analysis. For the data presented in Fig. 5A and B, the authors should quantify the GFP-BAF fluorescence intensity at the rupture site and in the nucleoplasm, and the nuclear envelope adjacent to the rupture site (or the GFP-BAS intensity at the nuclear envelope as a function of the distance from the rupture site). For the Zmpst24 depletion by siRNA (Suppl. Figure S2), the authors should quantify the immunofluorescence labeling and compared to siControl.

  • The effectiveness of the FTI treatment to facilitate recruitment of normally farnesylated lamins to the site of nuclear envelope rupture is impressive. To be able to better assess these results and to confirm the efficacy of the FTI treatment, i.e., to demonstrate what fraction of lamins remains farnesylated after 48 h of FTI treatment, the authors should include western blot data of lamin A or prelamin A levels following FTI treatment.

  • It is unclear what exactly was plotted in Figure 1D. In the main text, the authors refer to comparing the GFP-lamin A fluorescence intensity between the rupture site and an adjacent non-ruptured site at the nuclear envelope. However, the graph does not show any results for the adjacent site, and neither the figure legend nor the y-axis label refers to some normalization of the rupture site data to the non-rupture site data, as was done for example for panel 1F. The authors should clarify whether the data presented in panel 1D were normalized to a non-rupture site, which would be important to rule out that the gradual decline over time was caused by photobleaching during imaging, or whether the plot depicts only the data for the rupture site, in which case the data for the non-rupture site should be added.

Minor comments:

  • Figure 1. In the main text, the authors point to loss of lamin A/C fluorescence outside the cGAS-mCherry labeled rupture site. To support this statement, the authors should include the cGAS-mCherry color channel in the image series in Figure 1.

  • Fig. S1 is slightly confusing, since the top panel provides a schematic for the laser ablation (matching the experiments in the main Figure 1), whereas panels B and C report data from mechanical compression experiments and staining for endogenous lamins and BAF. The authors should consider moving the laser ablation schematic to the main figure and replacing it in the supplemental figure with a schematic of the mechanical compression assay.

  • Page 10, Section 2.5. The description of using “codon-optimized” version is confusing, since codon-optimized typically refers to adjustments for different species, whereas in these experiments both the host cells (BJ-5ta) and the expression constructs are human. Did the authors mean to say “siRNA resistant codons”? Please revise the text to clarify the meaning.

Reviewer 2 Report

Sears and Roux show the dependency of BAF and farnesylation for recruitment of lamins to rupture sites. Not much is known about the role of lamin accumulation in repair or stabilization of rupture sites. Sears and Roux combine spatially and temporally controlled laser-induced damage to the nuclear membrane with photobleaching and live imaging to follow the accumulation of tagged lamins at the site of damage. There use of farnesyl transfer inhibitors (and complementary RNAi experiments) and disease-related mutations in both lamin and the DNA crosslinking protein BAF provides new insight into the inhibitory role for farnesylation in mobilizing lamins to rupture sites as well further build on a key role for BAF as an upstream signal for lamin recruitment. Their powerful technique is currently the only way that this reviewer is aware of to get such detailed temporal resolution of rupture site protein accumulations.

The authors data mainly support their conclusions but I did find some issues with data representation and the text that would strengthen their manuscript.  

Major points:

  1. From what I was able to find, the b for each experiment is not provided in the main figure or the figure legend. Even if they have only one example, this information should be clearly provided. This is important because sometimes the laser induced puncture looks less severe than others (smaller holes perhaps) which could change the kinetics of repair and thus the accumulation of proteins to these sites. With this in mind, the approximate range of the rupture size should be noted in the main text or methods.
  2. The figure legends read like material and methods and don’t provide the pertinent minimal information necessary to understand the figure.
  3. The text and figure labels should all be carefully looked at carefully. The text is very confusing at times and there are many seemingly little mistakes that make the text very difficult to understand at times.

Specific points.

  1. Figure 1D is a quantitation of images in Figure 1B but follows figure 1C. It would be good to show the quantitation directly after the image.
  2. Figure 1C should be labeled that there is bleached nuclear pool. Also arrowheads need to be adjusted to label rupture site.
  3. Main text Lines 138-148 to describe Figure 1E-F is written in a confusing way that makes this complicated experiment difficult to understand. At some point unbleached is written when the authors likely meant bleached. Labels in 1F and 1E should be the same (eg “2 min post bleached half)). The colors in the cropped image in F are also confusing because they don’t match the bar graphs.
  4. In Fig. 4D it is not labeled that the experiment is with endogenous lamin A RNAi. The result of the delta 50 construct is confusing (if it is farnesylated why is it able to accumulate?). The conclusions made from this experiment (lines 265-271) need to be revisited.
  5. Line 371. “that interacts inhibits” should be corrected.

Round 2

Reviewer 1 Report

The authors have submitted a revised manuscript that now includes additional quantification and analysis, addressing the reviewers’ previous comments and resulting in an improved manuscript.